# Consumption of Substances in Nightlife Settings: A Qualitative Approach in Young Andalusians (Spain)

**DOI:** 10.3390/ijerph17165646

**Published:** 2020-08-05

**Authors:** María Ángeles García-Carpintero-Muñoz, Lorena Tarriño-Concejero, Rocío de Diego-Cordero

**Affiliations:** 1Research Group PAIDI-CTS 1050 Complex Care, Chronicity and Health Outcomes, Faculty of Nursing, Physiotherapy and Podiatry, University of Seville, 41009 Seville, Spain; agcarpin@us.es; 2Research Group CTS 969 Innovation in HealthCare and Social Determinants of Health, Faculty of Nursing, Physiotherapy and Podiatry, University of Seville, 41009 Seville, Spain; rdediego2@us.es

**Keywords:** adolescence, youth risk behavior, risk factors, substance addiction, use of nightlife venues, evidence-based prevention, qualitative research

## Abstract

Adolescence and youth are stages of exploration and experimentation, when the consumption of psychoactive substances for recreational or experimental purposes often begins. The general objective of this study was to explore youth consumption habits in nightlife settings and associated factors in Andalusia (Spain). To this end, we took into account young people’s perceptions about patterns of drug polyconsumption in nightlife settings and the perceptions and actions of health and teaching professionals towards this issue. We carried out a qualitative methodology with 24 in-depth interviews and 3 discussion groups with Andalusian girls and boys aged between 16 and 22 (n = 45) and 13 in-depth interviews with social agents (health and teaching professionals). We performed narrative discourse analysis and triangulation of identified categories and measured the units of analysis. The results show information relating to gender, age of initiation, most commonly consumed substances, motivation and effects, peer group pressure and how they obtained the substances, and the perceptions held and main activities carried out in the educational institutions and health centers.

## 1. Introduction

Adolescence and youth are exploratory stages in life characterized by a wide range of biological, psychological and social transformations, many of which generate conflicts, crises and contradictions [1]. The United Nations defines youth as the age cohort between 15 and 24 years, and within this, the first stage from 15 to 19 years, is called adolescence [2]. Although it is difficult to delimit where these terms begin and end, in view of the multiple socio-cultural diversities which exist [3], it is essentially a flexible concept whose meaning is constructed and reformulated according to the cultural, political and economic situation in each place and historical period [4].

It is important to point out that it is an exploratory stage, characterized by an increase in risk behavior, sensation-seeking and risk-taking [5,6] and experimental consumption of legal and illegal psychoactive substances, which link to certain habits and lifestyles, and have a major impact on an individual’s future health status [7,8].

We identified three levels of organization (individual, social and cultural) in relation to what motivates young people to follow risky consumption behavior. At the individual level, it is relevant to note that “sensation-seeking” is a personality trait defined by the need for varied, novel experiences and sensations [9], and the eagerness to take physical and social risks, and to accept the risk associated with any likely negative outcomes [10]. At a social level, there are the emotional stimuli of social interaction, the feeling of living “for the moment”, in which sex, alcohol and drugs are all “part of the fun” [11]. In the extensive, ephemeral sociability of adolescents’ shared leisure time, they play down risks, and everything revolves around the group: they go out, socialize, take drugs and experience the rites of passage to adulthood in a group [12]. By ‘youth nightlife’, we mean when young people and adolescents go for a night out, to a nightclub, disco or a bar, usually in the company of friends, or meet up to drink or take drugs in other areas such as parks, fields, abandoned lots or even a friend’s house [13]. This need to belong and feel part of a group is at its most intense in the stages of adolescence and youth, where individual and collective identity is formed [14,15,16]. On a cultural level, it mainly takes place on weekends, which are geared towards play and hedonistic pleasures [17]. Studies of youth culture have shown how leisure time has become the new measure of happiness in Western culture, and youths in particular understand it as an essential part of a desirable lifestyle [18]. For adolescents, nightlife equates to a break from the everyday routine, and a sense of freedom [12].

These individual, social and cultural elements explain how, in the context of adolescent leisure time, there is no single cause, but rather a combination of factors which leads to risky behavior, a mix of euphoria and collective excitement [11,15,17], where alcohol and drugs are a common denominator, which facilitates disinhibition and relationships between individuals, and in turn promotes risky situations [19].

Alcohol consumption is among the leading public health problems worldwide. Drinking alcohol is associated with a risk of developing health problems such as mental and behavioral disorders, including alcohol dependence, major non-communicable diseases, some cancers and cardiovascular diseases, among others [20]. Initiation to drinking alcohol, in societies where it is part of their culture, is universal. Drinking is used to mark social rites of passage of all kinds, including access to adult life, but it is also traditionally associated with learning self-control. However, in Western European countries, the mode of consumption is changing, with the globalization of products and patterns: young people from Anglo-Saxon countries have taken on the daily consumption of wine, so typical of Mediterranean countries, while young people from Southern Europe have started to consume more drinks with a high alcohol content and to binge drink [21,22]. Alcohol has become the universal fuel of the nightlife economy and its consumption is considered pleasant, fun and sociable [21,23,24,25].

When we consider the consumption of psychoactive substances, there are different degrees of social acceptability depending the type and intention—in other words, how dangerous the drug is considered, and where it is consumed. Heroin is considered the most marginal and stigmatizing, cocaine is seen as attractive, pills dangerous, cannabis natural and innocuous, and alcohol is the most socially accepted and omnipresent [6,26]. Consumption that takes place in leisure time and for recreational purposes, seeking fun and pleasure, is considered more acceptable [27,28].

In addition, drug polyconsumption is occurring, which is defined as the use of two or more psychoactive substances at the same time [29]. Some authors have established three modes of drug polyconsumption: pattern A (alcohol and tobacco), pattern B (cannabis with alcohol and/or tobacco) and pattern C (cannabis with alcohol and tobacco and at least one other illegal drug) [30]. These last two patterns increase the risks, enhance the effects of some substances on others, interfere with diagnosis and make treatment more difficult [29]. A previous intervention in 3882 Spanish adolescents pointed out that, in the short term, the consumption of more than one substance led to a higher probability of consuming other illegal substances, risky or abusive consumption of alcohol (binge drinking), traffic accidents, injuries, risky sexual relations with a greater probability of sexually transmitted diseases, fights, problematic use of the Internet or cyberbullying [31], criminal behavior and other acts of violence [32,33,34]. In terms of medium- and long-term consequences, it is associated with physical problems, low cognitive performance, depression and suicide. To make matters worse, the more substances consumed, the worse the clinical outlook and the greater the dependence [30,35].

At a global level, there are major differences in consumption patterns. The latest report on drugs stated that consumption among young people differs from one country to another according to social and economic circumstances, with two extreme categories: first, in high-income countries, where consumption occurs at rave parties, university cafeterias, concerts, and in recreational contexts to enhance sensations and experiences, using ecstasy, methamphetamines, cocaine, ketamine, LSD and gamma hydroxybutyrate (GHB), among others; secondly, in countries and areas with a low socioeconomic level, it is linked to young people living in marginal conditions, who consume drugs to escape from the harshness of their circumstances, and the substances most consumed are solvents, gasoline, paint, correction fluid and glue [36].

In Europe, addictive drugs are those that are capable of inducing habit, tolerance and physical dependence, where the doses need to be increased to achieve the same effect and whose deprivation produces a specific, manifest, measurable and observable syndrome, with harmful consequences for the individual and society [35]. Although there is no consensus over the legality or illegality of each substance in the legal systems of the different countries, studies are unanimous in including tobacco and alcohol, because of the habit and addiction they produce, even though they are mostly considered legal substances [37]. The data indicate that the most prevalent illegal substance is cannabis, followed by cocaine, 3,4-metilendioxi-metanfetamina (MDMA) and amphetamine [35]. In terms of gender, men have used these drugs over twice as much in the past year as women, with more intense and regular patterns. In terms of age, the prevalence is highest among young adults aged 15–34 [35], the onset of substance abuse is most pronounced in the period of early (12–14) and late (15–17) adolescence, and substance abuse peaks among persons aged 18–25 [36].

In Spain, the drugs most consumed by young people between 14 and 18 years old are alcohol and tobacco, followed by cannabis and sedative-hypnotics, with or without a doctor’s prescription. The ESTUDES Survey [38] showed that in late adolescence, there is a normalization and generalization of the abusive consumption of alcohol in both sexes, associated with weekend leisure (Spanish Observatory on Drugs and Addictions [38]). The average age of initiation into consumption is 13–16 years old, with no significant differences by gender. Consumption is “generalized” by type of substance, with women opting more for legal substances (alcohol, tobacco and sedative-hypnotics), and men for illegal ones [38], with an increasing consumption of a range of new psychoactive substances, such as spice, ketamine, ayahuasca, mephedrone, psychedelic sage (*Salvia divinorum*), hallucinogens, volatile inhalants, heroin, magic mushrooms, ecstasy, amphetamines and methamphetamines [29].

Andalusia follows the same pattern as Spain in terms of initiation, age, gender and type of substance consumed. However, there is a higher percentage of male polyconsumers, with the highest rates among users of heroin (average of eight substances), gamma hydroxybutyrate (GHB) (7.5), Ketamine (6.5) and magic mushrooms (6.1) [39].

As discussed above, substance use among adolescents leads to negative outcomes and risk behavior. Here, several studies pointed to the need to identify and respond early to reduce the associated risks in adolescence and adulthood [40,41,42]. For instance, Levy et al. 2016 stated that there needs to be an integrated public health approach to Screening, Brief Intervention and Referral to Treatment (SBIRT), involving not only health professionals but also educational institutions [43,44,45].

To date, various studies of substance consumption in adolescents and/or young people associated with risk factors have stressed the importance of detection and intervention programs. However, there is no research that jointly addresses the polyconsumption habits of young people and adolescents in nightlife settings and the perception of social agents related to education and health, which warrants the need to carry out this research using a qualitative methodology.

This study therefore aimed to explore youth drug consumption habits in nightlife settings and associated factors in Andalusia (Spain), taking into account the perception of young people regarding polyconsumption patterns in nightlife settings and the perceptions and action of health and education professionals to deal with this problem.

## 2. Materials and Methods

### 2.1. Design

We adopted a qualitative exploratory and descriptive design with a phenomenological approach [46]. The study also followed an ethnographic approach, since this allowed us to explore a particular topic in a specific context and in a specific subgroup [47]. This approach is characterized by (a) a conceptual orientation provided by a team of researchers; (b) a focus on a discrete community or group; (c) a focus on a problem within a specific context; (d) a limited number of participants; (e) the use of participants who may have specific knowledge; and (f) limited observation of participants [48].

### 2.2. Sample/Participants

The total number of participants was 58. We set up 24 interviews and 3 discussion groups (DG) (*n* = 21) with young people between the ages of 16 and 22, from the Andalusian provinces of Granada and Seville (Andalusia, Spain). In addition, we conducted 13 interviews with educational staff, promotion and prevention workers and healthcare employees.

For the selection of the sample, we used intentional sampling, through a process of searching for independent networks in different contexts to maximize the capture of different experiences, as well as to achieve greater control over bias in the selection of participants.

The inclusion criteria for the sample of young people were being aged between 16 and 22, with the following profile: high school or university students, consumers of alcohol and attending nightlife venues frequented by young people. As for the social agents, they had to have permanent and direct contact with young consumers in the fields of health and education.

### 2.3. Recruitment Process

Researchers from the anonymous project run in schools in Granada and Seville in 2019 carried out the semi-structured interviews with the young people. To recruit participants for the discussion group (DG) and agent key group, the 3 researchers from the team contacted professionals from the different areas involved, and, after informing them about the study, interviewed them.

### 2.4. Data Collection

For the interviews and DG with the young people, we used an interview script, structured in two thematic blocks: *Recreational and leisure environment* and *Recreational drug use patterns* (Table 1).

For the interviews with the key actors, we followed another interview script which included specific questions from the areas to be explored: education, health care and health promotion (Table 2).

We collected sociodemographic variables, such as environment, gender and age, to ensure the heterogeneity of profiles.

In addition, we held discussion groups to explore the issues raised in the interviews. This allowed us to find out the motivating factors that were expressed only in the group dialogue, and to establish contextual interpretations for the responses registered [49].

A final report was prepared with the statements of interview and DG informants, which is coded as follows: R (respondent); HC (health care employees); EE (educational employees); HPE (promotion health employers)-number. The analytical process was performed using the QSR NVivo 12 program.

### 2.5. Data Analysis

The interviews lasted on average 60 min and the DG, 45 min, and we recorded and transcribed both. We performed a narrative analysis of discourse by coding and categorizing the themes emerging from the interviews and DG, read the scripts, and made a first identification of codes and categories. After this first phase, another member of the research team carried out a triangulation of the identified categories.

After repeated readings of the transcripts, we identified the units of analysis and identified and grouped the categories, maintaining criteria of reflexivity and flexibility throughout the process. This whole analytical process was performed using the QSR NVivo 12 program. We completed the data collection process following the principle of theoretical saturation. 

### 2.6. Validity

This research followed the criteria of the Consolidated Criteria for Reporting Qualitative Studies (COREQ). The methods used to ensure validity were the triangulation of data, including participants with different socio-demographic characteristics, and the triangulation of data analysis through different researchers (Appendix A).

### 2.7. Ethical Considerations

The study was approved by the Research Ethics Committee of the University of Granada (CEI-UGR/883). All persons participating in the project voluntarily agreed to be interviewed, received information about the study with a letter from the research team guaranteeing confidentiality, and signed the informed consent form. The participation of minors also required the signature of their legal representatives in each case.

## 3. Results

### 3.1. Participants

The sample consisted of 45 adolescents (41.6% males and 58.4% females). Twenty-three were resident in Granada and 22 in Seville. The average age was 17.7 years old. All had consumed alcohol and/or other drugs/substances in nightlife settings. There were five educational employees (40% men and 60% women) with an average age of 36.2 years, who were working at that time in secondary schools, vocational training (*n* = 4) or universities (*n* = 1). There were also four promotion and prevention employers, who carried out specific programs with adolescents and young people in educational centers, specifically the Forma Joven Program, (25% men and 75% women), with an average age of 55 years, who taught the Forma Joven Program in middle schools, from the primary care network of the Autonomous Government of Andalusia, or from the Health Promotion Unit of Seville City Council. In the case of the healthcare workers, the sample was composed of two men and two women (50% men and 50% women), with an average age of 39.7 years, who worked at the Critical Care and Emergency Unit (DCCU) and in primary care for the Public Health Service of Andalusia (Spain).

### 3.2. Motivation. What Are They Looking for When They Consume?

Among the motivating factors for alcohol, tobacco and cannabis use, young people stress experimentation as the main reason to start consuming, as well as for pleasure, meeting new people, relaxing and escaping from problems (37 participants). We recorded the following statements:

R8: *“I’d say that drugs, apart from freeing you from your inhibitions and problems, make you feel superior, they make you feel like you’re God.”* (R21, R23)

R14: *“There, all my mates who usually meet up smoke—we all smoke. We started smoking together and it’s not like I’m going to get addicted, no. One of my mates comes along one day, we get together, he always rolls a joint and that’s it.”* (R22)

Educational professionals report that young people see consumption as a way of escaping or relaxing, or simply of integrating into a group.

EE2: *“At times in class when the subject comes up, they say they smoke joints because it relaxes them.”*

EE5: *“They do it to be accepted by the group, we see that at school too.”*

### 3.3. Polyconsumption. Alcohol, Tobacco, Cannabis… and What Else?

Among the responses reported by young people, there is evidence of a variety of substances and medicines consumed, from the commonest triad of alcohol, tobacco and cannabis, to the use of medicines or other drugs such as MDMA. Most of the participants (31 interviewees) admitted to having consumed cannabis, resins and oils derived from marijuana, and designer drugs, obtained on the illegal market. The following comments were recorded:

R29: *“The substance young people consume most, after alcohol and tobacco, is cannabis. To a lesser extent, pills and cocaine.”*

R3: *“So much cocaine, joints, MDMA, pills… pot, GHB (…). The liquid ecstasy thing I told you about. (…) I mixed it with more drugs. I mixed it that night with MDMA, with Speed, with cocaine… when I went out I didn’t just use one drug, maybe I mixed several.”* (R3, R4a, R4b, R5a, R12, R13, R19, R24).

R41: *“It’s a pill that makes your mouth really dry and makes you quite dizzy, because as people don’t buy alcoholic drinks, but they go there to take the drugs, the clubs are now forced to sell bottles of water for twelve euros.”*

On the other hand, educational professionals point out that, in school, the substances most consumed are tobacco, hashish and cannabis at break and when classes change over. The same substances are also mentioned by health promotion employers, with the addition of pills.

EE1: *“The most common substance is cannabis, consumed at breaks or outside the classroom (…) they talk to each other before starting the class, and on some occasions I’ve heard how they pool their money to buy alcohol, and even a kind of powder.”*

EE2: *“Tobacco and hashish in the breaks, and when they come in and out of the school.”*

HPE1: *“Alcohol, cannabis, marijuana and pills.”*

These can also be drugs bought in pharmacies, according to several of our informants:

R6: *“Do you know what it is? A cough syrup that has codeine in it, which is something, let’s see, how can I tell you… it has a substance that’s also found in heroin. And that’s in the cough syrup to sooth your cough. You mix that with Sprite and you drink it with ice, and it gives you a high like a reefer, and it tastes good too.”* (R5b, R8.)

Likewise, the most commonly used drugs, according to the four health care staff, were tobacco, alcohol, cannabis, cocaine and synthetic drugs:

HC1: *“I couldn’t tell what the typical profile is: we deal with young polyconsumers with very diverse characteristics, although they tend to be young people with a low level of purchasing power and culture and who tend to come from unstructured families.”*

HC2: *“I don’t think there’s a typical profile; many young people today are polyconsumers without seeming that way.”*

Respondents differentiate between “safe” and “addictive” consumption.

R16: *“I can go out without having to smoke—like playing, going to training, or going to a party and not smoking. It’s not something I depend on. I like doing it, but I’m not addicted to it. In fact, people consider marijuana a drug, I don’t consider it as a drug because tobacco is a drug too.”*

R19: *“Who doesn’t like temptation? But I control myself, I know that it’s bad for me and I don’t do it.”*

However, others mentioned that experimental consumption can turn into a habit:

R32: *“For me, marijuana is something that I think will be around for the rest of my life, I don’t know, I can’t deny it.”*

### 3.4. Gender. Does It Influence Polyconsumption?

The young people interviewed were not unanimous in their opinion about gender-related differences in consumption.

R2: *“Boys smoke more cigarettes and girls more joints.”* (R1, R4)

R35: *“It doesn’t matter if you’re a woman or a man. It could be that it affects women more… you know what I mean, it’s more or less similar, but for each person, each substance is different.”*

Here, educational professionals did not perceive a difference in relation to gender. Their opinion was shaped by anecdotes from girls about drug consumption after the weekend.

EE2: *“The consumption we can see here in the school is not a reflection of what they may have taken at the weekend or at night.”*

EE4: *“I’m not sure… they don’t usually tell us what they do when they speak in class, but then, first thing Monday morning is crazy-as they see you as a young person, they tell you about it, and I can sometimes hardly believe what I hear.”*

### 3.5. How Do They Obtain the Substances?

Among the young people interviewed, sales and distribution happen through a network of “contacts” known to them. Sometimes the young people grow drugs for their own consumption and sell the surplus to get some extra money.

We observed in this study that it is relatively easy to obtain substances of all kinds. For pharmacological substances, they go to pharmacies, and we have found one informant who can obtain it in his family environment, since one of his parents is a consumer.

R7b: *“Not other things, but unfortunately drugs are everywhere and you can get whatever you want.”*

R22: *“If I have weed and I have it to spare, I sell it and make a bit of cash.”*

R2: *“There are certain people we know who are the typical pushers, the ones who always sell everything, and yes, they drop by.”* (R18, R7a, R24)

In the educational field, it is difficult to obtain information about this, but some of our informants report that they have heard students talking in class about how they are going to get drugs for a party.

EE4: *“As the class starts or when you leave, a typical group forms, and they talk about people they buy from”… I heard once, “Go on, send him a WhatsApp and get him to save us some.”*

EE5: *“They have several, and I once heard “Well, if X doesn’t have any, we’ll get the car and go to Y.” Another one answered, “Yes but that one’s more expensive and I’m broke.””*

Health promotion professionals agree with the young participants about the usual way of obtaining substances, and say they get them through friends, family and drug dealers.

EE3: *“Colleagues and drug pushers.”*

EE5: *“Friends and family.”*

### 3.6. Places. Who Do They Do It with and Where?

As for the places where they consume drugs, it is usually in parks, outdoors or in bars, or sometimes at a friend’s house.

R3: *“We’ve been smoking in a park or quietly at someone’s house.”*

R36: *“For example, where I work…the people…in the bathroom we’ve seen little bags, so I asked my workmates what they were and they said “Well, that’s the people, they sometimes come here to the bars and take drugs..” Because once I saw an older girl having a beer who was falling asleep like that and I said “What’s up with this one?” Then they said “She’s been to the bathroom…” and I was like, “OK, right!”*

We also wanted to know the opinion of the health care employees about the places where they administer health care, whether in the public sector, at private homes or in emergency wards:

HC1: *“We see these young people mainly at homes, or in the street.”*

HC2: *“In my case, I see them at the health center.”*

The lack of any health care attention at these nightlife settings was also noted, as opposed to other options in other Spanish cities: 

HC1: *“We don’t go there, but I don’t think it’s necessary.”*

HC2: *“There isn’t, but it would be a good idea.”*

HC3: *“No, we don’t go there, we’re at the emergency wards and that’s where they usually are. In other cities like Ibiza, there are health staff at the clubs, but not here yet.”*

Educational staff point out that Thursday-night parties have become increasingly popular and the students usually buy the tickets before, which comes with a free drink. They also mention that consumption increases at the weekend.

EE1: *“On Friday when I leave work in the afternoon-evening, there’s an open field nearby, and they’re in the car with music and drinking, the other days of the week I don’t see them, some of them even go to our school.”*

EE3: *“As the weekend approaches, consumption seems to increase, and their attitude is usually a bit more laid back/slower.”*

EE5: *“Here at college we hear about university parties on Thursdays and Fridays. There are students who sell tickets with a free drink.”*

EE4: *“Especially on Thursdays, there are parties that they organize themselves… They talk about the ‘botellones’ (large groups of young people drinking in public places) and what each one is going to buy when the weekend comes.”* (EE10, EE11).

### 3.7. Effects. Unpleasant Experiences and Negative Effects.

In our interviews with young people, there was an association between polyconsumption and violence, mainly attributed to alcohol, as well as undesired effects and psychic symptoms, sometimes requiring treatment from health care givers.

R11: *“The most I’ve ever smoked was thirty joints. But I dropped dead that day.”* (R2a)

R13: *“I tried it in cakes and things like that, but it was too strong, so I don’t want to try it anymore… I remember being in bed, because I was in my house, lying on the bed as if I was asleep but I couldn’t react, although I could hear everything. I don’t know, it was just too much, so I don’t want to do it anymore.”*

R9: *“Mix any drug with alcohol, and all hell breaks loose.”* (R2b)

R27: *“When you take x substances what it does is… it doubles or triples whatever you feel… everything, so, it doubles or triples the violence, doubles or triples your sex-drive, but alcohol already does that, doesn’t it? That’s another drug, the only thing that it’s legal (Laughs). But these substances do it to you too.”*

Here, key health promotion actors in the field (four participants) consider that when short-term adverse effects arise, they primarily turn to friends rather than to health professionals. The adolescents and youths we interviewed are aware of the risks of substance use and the possible consequences of addiction, but the peer pressure from the group of friends is stronger than their individual decision.

HP2: *“They turn to friends, older relatives, and finally their parents.”*

HP3: *“They know the risks, but they follow the tribe.”*

Health care workers point out negative experiences they have experienced during health care in emergency situations.

HC1: *“Care usually consists of monitoring, knowledge of the clinical situation, stabilization, and assessment whether to transfer them to hospital.”*

Once the emergency is over, the agents point out that there is no protocol to follow up these young people:

HC3: *“It’s hardly ever done… it’s always through primary care consultations, and in more serious cases, through Detoxification Centers.”*

HC4: *“No follow-up, an emergency is dealt with, and that’s it.”*

### 3.8. Promotion, Prevention and Action

To avoid these situations, the healthcare workers interviewed point out that healthcare centers take some measures to prevent polyconsumption (for example through information campaigns, interventions with the population, among others).

HC2: *“[We run] workshops in schools, with the emphasis on primary care consultation.”*

HC4: *“Information on posters.”*

In the educational field, there is a notable lack of education and training expressed by these professionals to address this problem. Some of them refer to training acquired from previous experiences or through personal interest.

EE1: *“I have no training”* (EE4–EE8; EE10)

EE3: *“None. The only training I’ve had is my experience in life”*

They also refer to the lack of prevention programs aimed at students and that the educational institution has few protocols for action.

EE1: *“There’s no specific plan. The only things in the Regulations are the type of punishments for consuming drugs in the school (the sanctions model).”*

EE5: *“We have no prevention programs: the problem’s just addressed by dealing with cross-curricular subjects such as healthy living habits, physical exercise, anatomy or topics which lend themselves to making comparisons between healthy and harmful substances due to addiction, etc.”*

Health promotion professionals in the area of health have a higher level of training, and these workers claim to have received specific training courses given in continuous education courses by the District Health Organization.

HP1: *“I get training from my health district training unit.”*

HP4: *“Courses, scientific publications… I’m qualified in the Methadone Program, ‘Proyecto Hombre.”* (‘Proyecto Hombre’ is a project set up in Spain in 1984 which tackles addiction, from prevention to treatment, rehabilitation and reinsertion into society/work. https://proyectohombre.es/quienes-somos/#:~:text=Proyecto%20Hombre,as%C3%AD%20como%20a%20sus%20familias.&text=Tambi%C3%A9n%20trabaja%20en%20la%20prevenci%C3%B3n,de%20un%20mill%C3%B3n%20de%20personas.)

For more information on the themes and distribution of verbatim quotations, please consult Table A1.

## 4. Discussion

The objective of this study was to explore the consumption habits of young people in nightlife settings and associated factors in Andalusia (Spain), through the perception of young people and professionals working at educational and health centers, as well as the main actions carried out.

The main results shown are that the use of substances was not related to gender, and that the main motivation factor for substance use was for pleasure, escaping from problems, relaxation and integration into the peer group. The most commonly used substances were tobacco, alcohol and cannabis, and their use was sometimes linked with unpleasant experiences and negative effects, such as violence or psychological symptoms. The setting in which they consume ranges from parks or bars to their own homes. The substances are sold and distributed via word of mouth and at key places known to the peer group.

The professionals generally coincide with the adolescents interviewed in most aspects. It is noteworthy that very few prevention programs are run in the areas of education, and when they are, it is thanks to the health professionals associated with the ‘Forma Joven’ program at the request of the school or institution. We also found little training to address this issue taking place in the educational sphere, which is not the case with health professionals, who receive continuous training in this respect.

In relation to the influence of the gender variable, several studies highlighted that there are no significant differences between men and women in the consumption of legal substances such as alcohol and tobacco [31,50,51,52], but the percentage of consumption of cannabis and other drugs (ecstasy, amphetamines or hallucinogens) is higher in men [31,52,53].

Previous research linked changes in gender roles and the consumption of certain substances; in the case of alcohol, there may be a direct relationship between these social changes in young people and alcohol consumption [21,53,54,55]. Nevertheless, evidence from previous research reported greater polyconsumption in young men than in women [56]. Fernandez et al. (2018) stated that adolescents construct their masculinity through alcohol abuse, and that violence is a trait associated with “proving masculinity”. The interviewees share this idea, linking excessive alcohol consumption and violence and relate them to masculinity [57]. However, despite teachers referring to a higher consumption among young men (as reported by them), this figure may be biased because young men are more likely to talk about their experiences. Males are more explicit in their responses because they understand that they can freely exercise activities related to nightlife, such as enjoying going out, partying, and alcohol and drug use, without these activities damaging their reputation or undermining their masculinity. The opposite happens with girls, who tend to take the stance and declare in their statements that “they are good girls”, who repress their desire to drink alcohol and get drunk at parties, so as not to damage their “feminine” image, which is why they also follow rules of legal consumption [21].

The motivation factors which drive consumption include feeling euphoric, having hallucinations and increasing self-esteem [52]. Low resistance to peer pressure appears to be a risk variable in consumption. Consumption behavior also seems to be influenced by different psychosocial aspects, such as feeling integrated and accepted by the group, or to facilitate socialization [58], a view which educational professionals share. Previous studies have pointed out that among the factors which influence the initiation and stabilization of substance use in youths, peer group pressure is paramount, since relationships with peers are more important in this period and they spend more time with their peer group. Thus, there is a general consensus over the idea that one of the most powerful predictors for substance use is being accompanied by friends who engage in this type of practice [58,59,60]. On the other hand, Romo-Avilés in 2015 identified that the peer group functions as a protective community which looks after its members’ safety when risky consumption takes place [55].

Polyconsumption has become another common practice. Among the possible combinations of psychoactive substances which characterize polydrug use, the most common according to several studies are alcohol, tobacco and cannabis [3,27,30,61]. Occasionally, in special situations, this may include other substances such as cocaine, ecstasy or designer drugs [30].

There is a clear distinction here between “safe” consumption and “addiction”. This idea coincides with the literature, in which the figure of a weekend consumer of legal and illegal drugs is considered perfectly natural and normalized and is associated with the typical recreational experimentation of young people [27], to such an extent that young teens often consider sporadic consumption harmless, which stresses the relationship between consumption and low risk perception [61].

Strategic points of supply and consumption exist in most cities where there are certain neighborhoods, usually found on the city outskirts, linked to marginal culture and with low socioeconomic levels, where substances are trafficked, as Ferrer pointed out in one of her interventions [62]. The players involved in this process are the traffickers, who supply the pushers, who sell to the consumers, as well as other intermediaries who perform support tasks (vigilance, raising the alarm if the police appear, etc.) [63]. The educational professionals confirm the existence of these figures, and that young people usually have their phone numbers so they can confirm the purchase before the event or simply to know where to pick it up.

This contrasts with other articles, which show that in the vast majority of cases, the sale, distribution and purchase of illegal substances occurs online, so Internet users are able to evade the authorities [64], and new buying and selling practices also occur using collective appointments made by phone [63].

For promotion, prevention and action, the study by Levy et al. 2016 stated that an integrated public health approach of Screening, Brief Intervention and Referral to Treatment (SBIRT) is needed [41], involving not only health professionals but also educational institutions [43,44,45]. We have included the figure of these social agents in our study because of their key role in identifying and responding early to reduce the risks in adolescence and adulthood, as reported in other studies conducted in the area of adolescent drug use [40,41]. As verified in this research, a successful SBIRT promotes behavioral changes by helping these adolescents to resolve the ambivalence of their changes in behavior, through empathetic interview styles, with guided debates about the perceived harm of drug use and the benefits of abandoning the use of substances [65].

In addition, our study, together with the one conducted by Kristen et al. 2019, shows that there are gaps in our knowledge in the educational field and our capacity to provide evidence-based prevention and early intervention. Most of the teachers had not received any kind of training in this area, and more training is certainly needed [41].

This study has certain limitations. We chose our sample on the basis of convenience, which makes it difficult to extrapolate wider conclusions from the results obtained. Nevertheless, the results can serve as a reliable approximation of the real situation, and the qualitative methodology used allowed for a precise approach to these issues. Another limitation is that we did not measure the previous information that the participants have about the topic, and this would have been interesting to include as a variable of the study in order to obtain more detailed results about the level of previous knowledge and the relationship with the answers obtained.

We believe that analyzing the discourse of young people, adolescents and social agents is an appropriate way to continue working on these issues to improve our knowledge and ways of addressing this social problem. This research can be considered as an approach to a contemporary reality which is of considerable relevance to social, political and health issues. We need to create a wide range of action protocols, and adolescents and young people, health professionals and teachers should work together to design evidence-based prevention and intervention programs.

## 5. Conclusions

In our study related to gender, the inclusion of young women as consumers in the same percentages as their male peers and beginning in adolescence will lead to a serious health problem in this decade. As for the types of drug consumed, we found that alcohol is omnipresent at all ages and types of consumption, and we have observed a “generalized” use of substances and their consumption. Women remain within the parameters of “legality” with poly-consumption of tobacco, alcohol and sedative-hypnotic drugs dispensed in pharmacies, without crossing the lines of legal risk. Meanwhile, in addition to legal drugs, if men risk accessing the illegal market to obtain other types of substances, the most commonly consumed drug is cannabis, but there are others such as cocaine, speed, MDMA, ecstasy or GHB. The ease of socializing, relaxation, getting euphoric and escaping from problems in a fun, party-like environment, generally at weekends, are the main motivating factors. Sometimes, undesirable effects such as drowsiness and distortion of perception appear, mainly related to excess consumption of cannabis, or aggressiveness related to the excessive consumption of alcohol or a mixture of this with other substances.

The peer group is one of the most influential factors in initiating and maintaining consumption. The sale or distribution of illegal substances occurs through known contacts at strategic points. On the other hand, drugs such as codeine, generally in the form of syrups mixed with alcohol, are available in pharmacies with or without a prescription.

In the area of health and education, health professionals provide more education, training and prevention, detection and intervention programs, whereas educational professionals do much less: teachers should be more involved in delivering such programs, since adolescents and young people spend a large part of their lives in educational institutions.

We believe that it is vital for teachers, health care workers and parents to continue working on these aspects through a qualitative approach with young people, adolescents and social agents to improve our knowledge of this social problem. We hope this research will help these professionals to understand more about a contemporary issue which has enormous relevance in social, political and health fields. They should plan new lines of action to address them, and adolescents and young people, health professionals and teachers alike should work collectively to design evidence-based prevention and intervention programs.

## Figures and Tables

**Table 1 ijerph-17-05646-t001:** Interviews and discussion group (DG) guide.

1. Sociodemographic data: age, gender, origin (environment/city), habitual consumption.
2. Themes *2.1. Recreational and leisure environment:* Preferred days for going out at night. Most common habits and settings chosen for nightlife. Models of organization within the group (relationship between peers). Preferences for venue (public or private), time schedules.
*2.2. Patterns of recreational drug use.* Substances consumed when out at night: types, quantity, frequency, consumption over the last month, how it is bought. Reasons for drug polyconsumption. Differences between genders. Expectations over consumption and related experiences.

**Table 2 ijerph-17-05646-t002:** Script for social informant interviews.

1. Sociodemographic data: age, gender, origin (environment/city), scope: health (health promotion and health care), and education.
2. Themes: *2.1. Approach from education* How does your educational institution (college, university) address the prevention of substance use (alcohol, tobacco and other drugs)? Describe the times when substance use by adolescents increases. What are the most common substances they use? What is the protocol to follow when you detect cases of drug consumption in the college or when students are seen to have consumed drugs? Are there particular days or times in the year when you detect that adolescents consume more? What usually happens those days? What training have you received to address this problem? Have you detected any contacts (e.g., contact person who supplies the drugs) in the institution, or outside it? How are these contacts made?
*2.2. Approach from health promotion* Is the motivation to consume drugs dealt with in the FORMA Joven ^1^ health promotion program? When are the drugs consumed (daily, during the week, at weekends)? What types of substances do they consume? What are the short-term adverse health effects? Who do they turn to if they suffer adverse effects? In the medium and long term, do they reflect on addiction and adverse health effects? Where do they go to buy the substances, and who do they buy them from?
*2.3. Approach from health care* What primary health care measures are taken to prevent drug polyconsumption, for example through awareness-raising campaigns, interventions with the target population, etc.? Are any kind of health workers present at nightlife settings? In cases of emergency: where are young people treated, what care is given to them, and how do they get there? In your experience, what is the typical profile of the young drug polyconsumer? What substances/drugs are most frequently detected? After emergency care, is there any kind of follow-up of these young people?

^1^ FORMA Joven’, is a Program of the Andalusian Regional Government, with strategies to promote healthy life habits among adolescents and young people. https://www.formajoven.org/.

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
