# Peer review of "Consumption of Substances in Nightlife Settings: A Qualitative Approach in Young Andalusians (Spain)"

_ijerph, 2020, doi:10.3390/ijerph17165646_

Round 1
Reviewer 1 Report
This is an interesting manuscript about a qualitative approach exploring the consumption of substances in nightlife settings among young Andalusians.
The introduction is excellent and the literature current. This section describes the life stage of adolescence comprehensively and from a refreshingly strengths-based perspective.
However, the reviewer is not presented with a full description of the kinds of night life settings in terms of location and times where the respondents were interviewed. It lacks a feel of the ethnographic experiences which the authors mention in the manuscript. The manuscript will be enhanced by a fuller description (144).
Also please spell out the names of the drugs GHB (95) and MDMA (106)
I was surprised to see the question about the assumption that drugs are bought. Adolescents can also secure them in other ways. An open-ended question more appropriate to qualitative approach would have been preferable. Stated simply: how do you acquire your drugs? The options would have been wider in term of responses but asking only how it is bought, stifles the options (170).
Since this is such a critical stage of life with much exploration, adventure and risk, it would have been helpful if implications were drawn which were connected to the findings, to inform, teachers and health care workers and others, like parents and the adolescents themselves, about preventive interventions. SBIRT which the authors reference is a great start. The implications need to be fleshed out for this research to be useful for practitioners.
Author Response
This is an interesting manuscript about a qualitative approach exploring the consumption of
substances in nightlife settings among young Andalusians. The introduction is excellent and the
literature current. This section describes the life stage of adolescence comprehensively and
from a refreshingly strengths-based perspective.
Response: Thank you for your kind words.
However, the reviewer is not presented with a full description of the kinds of night life settings in
terms of location and times where the respondents were interviewed. It lacks a feel of the
ethnographic experiences which the authors mention in the manuscript. The manuscript will be
enhanced by a fuller description (144).
Response: A specific description of leisure time is included in the text, with bibliography (144).
Also please spell out the names of the drugs GHB (95) and MDMA (106)
Response: GHB is Gamma Hydroxybutyrate and MMDA is 3,4-metilendioxi-metanfetamina.This
has been corrected in the text.
I was surprised to see the question about the assumption that drugs are bought. Adolescents
can also secure them in other ways. An open-ended question more appropriate to qualitative
approach would have been preferable. Stated simply: how do you acquire your drugs? The
options would have been wider in term of responses but asking only how it is bought, stifles the
options (170).
Response: We agree with you. Thank you for the observation. We consider this aspect for
future research.
Since this is such a critical stage of life with much exploration, adventure and risk, it would have
been helpful if implications were drawn which were connected to the findings, to inform,
teachers and health care workers and others, like parents and the adolescents themselves,
about preventive interventions. SBIRT which the authors reference is a great start. The
implications need to be fleshed out for this research to be useful for practitioners.
Response: Thank you for the observation. We have included a paragraph related to the
implications and its corresponding bibliographic reference [67].
Also during the review process, we noticed that the correspondence author's affiliation was not
at the beginning of the text and we added it.
Furthermore, the language has been revised again by the same specialized translator.

Reviewer 2 Report
Thank you for the opportunity to review this manuscript. Strengths include an up-to-date look at young people’s experience of substance use in Spain including the startling revelations of dangerous behavior in the youths' direct quotes. Weaknesses include a lack of any quantitative data, even though quantitative comparisons are implied; and some issues of organization and clarity.
36 levels of what? (sentence needs to be reworded, also use active voice to be more straightforward & concise;)
43/48 not sure how the word “relational” differs from “social” to be a different category. Maybe the word “social” is a better word for your description of “relational” and “cultural” or even “societal” is a better word for what you are describing as “social.” The word “social” connotes direct relationships and interactions between people, whereas “societal” refers to the culture or climate of a place and time in history.
71 “tolerance” do you mean social acceptability (use is tolerated by others), or physiological tolerance? It sounds like you are describing social perception
In the intro, can you talk more about the relevance of nightlife? Are you drawing a fundamental distinction between daytime/individual use and use in the context of going out with friends? How are you defining nightlife, exactly? Does it mean going to clubs with friends, or large parties? Would hanging out with one friend or date at night count as nightlife? Does using drugs alone in preparation for meeting up with friends count? How did you deal with participants who were using both in nightlife and in their individual daily lives? It seems like some of the interview questions weren’t limited to nightlife use.
Results
There are two section 3.2s.
Sections 3.2 through 3.6 seem really out of order conceptually (likewise the flowchart at 396). An order like
3.2 Motivation. What are they looking for when they consume?
3.3 Polyconsumption. Alcohol, tobacco, cannabis... and what else?
3.4 Gender. Does it influence polyconsumption?
3.5 How do they obtain the substances?
3.6 Places. Who do they do it with and where?
3.7 Effects. Unpleasant experiences and negative effects.
3.8 Promotion, Prevention and Action
might flow better
237 any numbers to clarify “main reason”?
251 any numbers to clarify “most”?
275 any numbers to clarify “most commonly used”?
302-307 seems to mix two different ideas (confusing): 1. Peer pressure; 2. Who to go to for help in a drug-related medical situation
303 any numbers to clarify “primarily”?
396 I’m not sure a flowchart, which suggests pathways, is the best choice for presenting your results. Your reference to the chart as a “map” implies directional relationships between the constructs, but the concepts here aren’t linked in that way. Maybe a table? Or, leave out the chart as it’s not particularly informative
430 any evidence that men are more likely to talk about their experiences than women?
479 what is meant by “global”?
479 what do you mean by “previous information that the participants have about the topic” being a limitation? Do you mean that your not measuring it is the limitation?
483 maybe elsewhere as well: what’s the distinction between “young people” and “adolescents” ?
483 the approach of discourse analysis (which is not used in the present study) appears abruptly at the end of the discussion. You are probably not intending to introduce a new approach at this point (as d.a. would really be for a different type of research purpose), either as a recommendation or as a reflection on the present study, so you probably need to reword this.
490-492 confusing sentence. Unclear what you are trying to say about gender. Why is drug use more worrying/more problematic in females than in males? Also, wording needs work (e.g., statistics can’t lead to a health problem; sentence is wordy/convoluted)
Writing
49 change “at” to “on”
50 change “in” to “of”
54 individual individuals
61 overly dramatic wording suggests annihilation of the species. Please be more specific about threats posed by alcohol, perhaps citing some of the stats from the WHO website you cite.
more concise by just saying, “… with a high alcohol content and to binge drink”)
Many instances of wordiness and passive voice which should be cleaned up for a more concise, straightforward manuscript. Probably any time you have the word “regard” (“as regards” “in/with regard to” etc) you can get rid of it and re-word more straightforwardly.
337 grammar and punctuation errors
370 space
425 incomplete comparison (greater polyconsumption than monoconsumption? Or greater than in women?)
452-455 very wordy sentence; convoluted
455-456 seems to contradict. Do the dealers sell directly to the buyers/users, or do the pushers?
479-482 wordy, convoluted sentence. Needs to be two, clearer sentences
504-507 two-sentence paragraph seems to lack a unifying purpose
533 under R18a (Places…) extra quote mark
Author Response
Thank you for the opportunity to review this manuscript. Strengths include an up-to-date look at
young people‟s experience of substance use in Spain including the startling revelations of
dangerous behavior in the youths' direct quotes. Weaknesses include a lack of any quantitative
data, even though quantitative comparisons are implied; and some issues of organization and
clarity.
Response: Thank you for the observation.
36 levels of what? (sentence needs to be reworded, also use active voice to be more
straightforward & concise;)
Response: This sentence has been modified, remaining as follows: We identified three levels of
organization (individual, social and cultural) in relation to what motivates young people to follow
risky consumption behavior.
43/48 not sure how the word “relational” differs from “social” to be a different category. Maybe
the word “social” is a better word for your description of “relational” and “cultural” or even
“societal” is a better word for what you are describing as “social.” The word “social” connotes
direct relationships and interactions between people, whereas “societal” refers to the culture or
climate of a place and time in history.
Response: Thank you for the observation. „Relational‟ has been changed to „social‟ and „social‟
to „cultural‟.
71 “tolerance” do you mean social acceptability (use is tolerated by others), or physiological
tolerance? It sounds like you are describing social perception.
Response: When we refer to „tolerance‟ we mean „social acceptance‟. This has been modified in
the text.
In the intro, can you talk more about the relevance of nightlife? Are you drawing a fundamental
distinction between daytime/individual use and use in the context of going out with friends? How
are you defining nightlife, exactly? Does it mean going to clubs with friends, or large parties?
Would hanging out with one friend or date at night count as nightlife? Does using drugs alone in
preparation for meeting up with friends count? How did you deal with participants who were
using both in nightlife and in their individual daily lives? It seems like some of the interview
questions weren‟t limited to nightlife use.
Response: By „youth nightlife‟, we mean going for a night out, to a disco or a cocktail bar,
usually in the company of friends. This time implies some special characteristics that define and
differentiate it from the time which is "not leisure time", which is a time of work or study, (51-54),
and we include bibliographic citation.
Results
There are two section 3.2s.
Sections 3.2 through 3.6 seem really out of order conceptually (likewise the flowchart at 396).
An order like
3.2 Motivation. What are they looking for when they consume?
3.3 Polyconsumption. Alcohol, tobacco, cannabis... and what else?
3.4 Gender. Does it influence polyconsumption?
3.5 How do they obtain the substances?
3.6 Places. Who do they do it with and where?
3.7 Effects. Unpleasant experiences and negative effects.
3.8 Promotion, Prevention and Action
might flow better
Response: Thank you for the observation: we agree with you. Actually, the numbering was a
mistake. We have now changed the order of the sections.
237 any numbers to clarify “main reason”?
Response: we have introduced that data in the text: 37 participants.
251 any numbers to clarify “most”?
Response: Clarified in the text: 31 interviewees
275 any numbers to clarify “most commonly used”?
Response: Added in text ‟the four health care staff‟.
302-307 seems to mix two different ideas (confusing): 1. Peer pressure; 2. Who to go to for help
in a drug-related medical situation.
Response: This has been clarified in the text.
303 any numbers to clarify “primarily”?
Response: The four health professionals mentioned above in the text.
396 I‟m not sure a flowchart, which suggests pathways, is the best choice for presenting your
results. Your reference to the chart as a “map” implies directional relationships between the
constructs, but the concepts here aren‟t linked in that way. Maybe a table? Or, leave out the
chart as it‟s not particularly informative.
Response: Thank you for the observation. The chart has been deleted.
430 any evidence that men are more likely to talk about their experiences than women?
Response: We have added these two sentences to the text: “Males are more explicit in their
responses because they understand that they can freely exercise activities related to nightlife,
such as enjoying going out, partying, and alcohol and drug use, without these activities
damaging their reputation or undermining their masculinity. The opposite happens with girls,
who tend to take the stance and declare in their statements that "they are good girls", who
repress their desire to drink alcohol and get drunk at parties, so as not to damage their
"feminine" image, which is why they also follow rules of legal consumption.” (434-440).
479 what is meant by “global”?
Response: The term „global‟ has been changed to „precise‟.
479 what do you mean by “previous information that the participants have about the topic” being
a limitation? Do you mean that your not measuring it is the limitation?
Response: This has been changed. We meant „Participants' prior information on drugs is not
measured‟.
483 maybe elsewhere as well: what‟s the distinction between “young people” and “adolescents”
? Response: The United Nations‟ delimitation between stages of “youth” and “adolescence” is
specified in the text (35)
483 the approach of discourse analysis (which is not used in the present study) appears
abruptly at the end of the discussion. You are probably not intending to introduce a new
approach at this point (as d.a. would really be for a different type of research purpose), either as
a recommendation or as a reflection on the present study, so you probably need to reword this.
Response: the sentence has been changed to “We believe it is vital for teachers, healthcare
workers and parents to continue working on these aspects through a qualitative approach with
young people, adolescents and social agents to improve our knowledge of this social problem.
490-492 confusing sentence. Unclear what you are trying to say about gender. Why is drug use
more worrying/more problematic in females than in males? Also, wording needs work (e.g.,
statistics can‟t lead to a health problem; sentence is wordy/convoluted). Response: This
sentence has been deleted from the text.
Writing
49 change “at” to “on”.
Response: This has been corrected.
50 change “in” to “of”
Response: This has been corrected.
54 individual individuals
Response: Please note that „individual‟ in this phrase is an adjective, referring (previously) to
the 3 types of elements: individual, relational and social. However, in light of the comment on
line 43/48, we have changed the sentence to „These individual, social and cultural elements…‟
(N.B. „individual‟ is still an adjective here)
61 overly dramatic wording suggests annihilation of the species. Please be more specific about
threats posed by alcohol, perhaps citing some of the stats from the WHO website you cite.
Response: This has been changed in the text.
more concise by just saying, “… with a high alcohol content and to binge drink”)
Response: This has been corrected in the text
Many instances of wordiness and passive voice which should be cleaned up for a more concise,
straightforward manuscript. Probably any time you have the word “regard” (“as regards” “in/with
regard to” etc) you can get rid of it and re-word more straightforwardly.
Response: A second language review has been done by Simon James Armour (BA Cambridge
University (1984); MA Cambridge University (1985); PGCE Cambridge University (1985); Head
of Translation at International House Language School, Cordoba, Spain.
337 grammar and punctuation errors.
Response: See comment above
370 space
Response: This has been corrected.
425 incomplete comparison (greater polyconsumption than monoconsumption? Or greater than
in women?)
Response: This has been corrected.
452-455 very wordy sentence; convoluted.
Response: See comment above on language review.
455-456 seems to contradict. Do the dealers sell directly to the buyers/users, or do the
pushers? Response: We have clarified this by reformulating to: “The players involved in this
process are the traffickers, who supply the pushers, who sell to the consumers, as well as other
intermediaries who perform support tasks”
479-482 wordy, convoluted sentence. Needs to be two, clearer sentences.
Response: We have corrected the paragraph
504-507 two-sentence paragraph seems to lack a unifying purpose.
Response: corrected
533 under R18a (Places…) extra quote mark
Response: This has been corrected.
Also during the review process, we noticed that the correspondence author's affiliation was not
at the beginning of the text and we added it.
Furthermore, the language has been revised again by the same specialized translator.
